# Constant Flux Layers with Gravitational Settling: with links to aerosols, fog and deposition velocities over water.

Peter A. Taylor[1]

[1]Centre for Research in Earth and Space Science, York University, Toronto, M3J 1P3, Canada

*Correspondence to*: Peter Taylor (pat@yorku.ca)

**Abstract.** Turbulent boundary layer concepts of constant flux layers and surface roughness lengths are extended to include the effects of gravitational settling. These impact vertical profiles and surface deposition of aerosols, including fog droplets, especially over water. Simple profile solutions are possible in neutral and stably stratified

atmospheric surface boundary layers.

**Keywords**  Constant flux layers •  Aerosols • Fog • Gravitational settling • Surface roughness

## 1. Introduction

Within the turbulent atmospheric "surface layer", typically $0 < z < \sim 50$ m, it is helpful to look at idealized situations where fluxes of momentum, heat or other quantities are considered independent of height, z, above a surface which is a source or sink of the quantity being diffused by the turbulence. Garratt (1992, Chapter 3) or Munn (1966, Chapter 9) discuss this "constant flux layer" concept and, for momentum, the paper by Calder (1939), discussing earlier work by Prandtl, Sutton and Ertel, is an early recognition of the utility of this idealized concept. Monin-

Obukhov Similarity Theory (MOST) is based on constant flux layer situations in steady state, horizontally homogeneous, turbulent atmospheric boundary layers and leads to suitably scaled, dimensionless velocity and other profiles being dependent on $z/L$ where z is height above the surface and $L$ is the Obukhov length (defined below). With no sources or sinks of momentum or heat within these constant flux layers one can use dimensional analysis to establish the form of the profiles while observational data or hypotheses are needed to establish the detailed profile

forms. Munn (1966, Chapter 9), Garratt (1992, section 3.3) or Kaimal and Finnigan (1994) explain Monin-Obukhov similarity while Monin and Obukhov (1954) is a translation of the original Russian work. The simplest case is with neutral stratification ($1/L = 0$) where dimensional analysis can be used to infer that the velocity shear, $dU/dz$ is simply proportional to $u_*/z$ where the shear stress, assumed constant with height, is $\rho u_*^2$, with $\rho$ as air density.

Integration of this relationship leads to

$$U(z) = (u_*/k) \ln(z/z_{0m}), \tag{1}$$

with the roughness length for momentum, $z_{0m}$, being defined as the height at which a measured profile has $U = 0$

when plotted on a $U$ vs ln z graph, and where $k$ is the Karman constant with a generally accepted value of 0.4.





Noting that $z_{0m}$ values are generally small compared to measurement heights, and after a $z_{0m}$ value has been established for the underlying surface, it is mathematically convenient to modify the relationship to

$$U = (u_*/k)\ln((z+z_{0m})/z_{0m}), \qquad (2)$$


so that we have $U = 0$ on $z = 0$. In eddy viscosity terms this corresponds to

$$K_m = ku_*(z+z_{0m}) \qquad (3)$$

In situations with constant, or near constant fluxes of heat ($H$) or water vapour, similar, near logarithmic, MOST profiles and eddy diffusivities can be established, based on measured profiles, involving $z/L$ where the Obukhov length, $L = -\rho c_p \theta u_*^3/(kgH)$ in which $c_p$ is the specific heat of air at constant pressure, $g$ is acceleration due to gravity and $\theta$ is the potential temperature. For potential temperature and water vapour profiles these can involve additional "scalar" roughness lengths, $z_{0h}$ and $z_{0v}$. Much has been written about roughness lengths and ratios between

$z_{0m}$ and $z_{0h}$, including Chapter 5 of Brutsaert (1982). For momentum transfers, pressure differences and form drag on roughness elements, sand grains, blades of grass, bushes, trees, buildings and water waves can provide most of the drag on the surface and, except over water, $z_{0m}$ is considered as a Reynolds number independent surface property. Water waves are wind speed dependent and $z_{0m}$ needs to take this into account. For heat and water vapour the final transfers from air to the surface involve molecular diffusion and, as a result, values of $z_{0h}$, $z_{0v}$ are generally lower

than $z_{0m}$. For aerosols and droplet concentrations we will introduce an additional roughness length, $z_{0c}$, on the basis that their interactions with the surface will again be different from other quantities. Aerosol type, density and size, as well as $u_*$, may also cause variability in $z_{0c}$. As was necessary with the established roughness lengths for momentum and heat, field measurements over a variety of surfaces will be needed to establish appropriate values. As a first approach, for fog droplets and other aerosol particles deposited to water surfaces we assume $Qc \to 0$ as $z \to 0$ and,

as a trial value, use $z_{0c} = 0.01$m.

**2. A simple model with added gravitational settling**

We will consider situations where there is aerosol present with a concentration or mass mixing ratio, Qc. For simplicity it is assumed to consist of uniform particles with a constant gravitational settling velocity, $V_g$, and is at a

density low enough to have no impact on the density of the combined air + aerosol mixture. We assume no mass exchange between the aerosol and the surrounding air, which may be a concern for fog droplets which require an additional assumption that the air is always at 100% relative humidity.

If we have a net upward or downward flux of aerosol we need to discuss the source. If we are considering sand or

dust being picked up from the surface by wind then upward diffusion will be countered by downward gravitational settling, while if the source of the aerosol is above our constant flux layer then the turbulent fluxes and gravitational settling combine. This could be the case with long range transport of aerosol in air blowing out over a lake or the





ocean. Our other example will be fog droplets, formed at the top of a fog layer and being deposited at the underlying surface.


In a horizontally homogeneous, steady state situation, and with a simply specified eddy diffusivity (Eq (3) but with $z_{0m}$ replaced by $z_{0c}$) and neutral stratification we just need to consider vertical turbulent transfers and gravitational settling. One could then model the constant downward flux of aerosol, $F_{Qc}$, as

$$V_g Qc + ku_*(z + z_{0c})\,dQc/dz = F_{Qc} = u_* q_{c*}, \tag{4}$$

where $V_g$ represents the gravitational settling velocity, proportional to $d^2$, where d is the diameter, via Stokes law for small (d $<$60 µm) spherical particles (Rogers and Yau,1976, p125), and $u_*$ is the friction velocity. We introduce $q_{c*}$ as a mixing ratio scale via this definition. The eddy diffusivity $K_{qc}$ is assumed to be

$$K_{qc} = ku_*(z + z_{0c}), \tag{5}$$

where $z_{0c}$ is a roughness length for the aerosol with the assumption that $Qc = Qc_{surf}$ at $z = 0$.

The upward flux case with a surface source of aerosol is interesting in the sense that there will only be a steady,

horizontally homogeneous, state when the net flux is zero, i.e, upward turbulent transfer is balanced by gravitational settling. Xiao and Taylor (2002), in an aside from a blowing snow study, show that this leads to the classic power law solution (e.g, Prandtl, 1952), which in the current context is

$$\ln(Qc(z)/Qc_{surf}) = -S\zeta, \text{ where } \zeta = \ln((z+z_{0c})/z_{0c}) \text{ and } S = V_g/(ku_*)$$

or
$$Qc(z) = Qc_{surf}\,((z+z_{0c})/z_{0c})^{-S} \tag{6}$$

Profiles of suspended sediment, and velocity, in water currents can be treated in a similar way but there is an

interesting twist if the density of the sediment and water mix is sufficient to modify the turbulent mixing through stable stratification. Taylor and Dyer (1977) rediscovered an interesting result due to Barenblatt (1953) showing that a modified solution allowing for stratification effects on the eddy diffusivity could be obtained. Observations were sometimes misinterpreted as power laws with a modified value of $k$ (Graf, 1971, p180).

For the downward flux case to the lower boundary in the atmospheric surface layer it is easiest if we assume $Qc_{surf} =$

0, which may be most relevant over water. Material starts from a source above the constant flux layer and travels downwards due to both turbulent mixing and gravitational settling. Assuming constant values for $z_{0c}$, $u_*$ and $V_g$ one can then solve the first order ODE, Eq (4), by integrating factor techniques. Multiplying Eq. (4) by $(z+z_{0c})^{S-1}/(ku_*)$ where $S = V_g/(ku_*)$, gives,




$$(d/dz)[(z+z_{0s})^S Qc] = (q_{c*}/k)(z+z_{0c})^{S-1} \qquad (7)$$

and, with $Qc(0) = 0$ the solution is,


$$Qc(z) = (q_{c*}/(kS)) \ [1- ((z+z_{0c})/z_{0c})^{-S}]. \qquad (8)$$

In terms of $\zeta = \ln ((z+z_{0c})/z_{0c})$, we can write,

$$Qc(\zeta) = (q_{c*}/(kS)) \ [1-e^{-S\zeta}]. \qquad (9)$$


These can be referred to as Constant Flux Layer with Gravitational Settling, CFLGS, profiles. In the limit as $V_g$ and $S \rightarrow 0$, and as $\zeta \rightarrow 0$, Eq (9) gives $Qc(\zeta) = (q_{c*}/k) \ \zeta$, a standard log profile.

**3. Dry deposition velocities**

For aerosol dry deposition (i.e. not involving rain or snow - wet deposition) to any surfaces the traditional way to parametrize the process is with a deposition velocity, $V_{dep}$. Then the flux to the surface is represented as,

$$F_{Qc} = V_{dep} \ Qc \ (z_{ref}). \qquad (10)$$

In a numerical model the reference height $z_{ref}$ is often the lowest grid level. If gravitational settling is the main cause of $F_{Qc}$, we would expect little change in $Qc$ with height, but if turbulent transfer is dominant then the choice of $z_{ref}$ could be important.

Dry deposition can involve many aspects and is often modelled in terms of a series of resistances. The deposition 135 velocity used generally includes the effects of both gravitational settling and turbulent collisions of particles with vegetation or the ground, or water surface. The expression used for deposition velocity by Zhang et al (2001), and others, is

$$V_{dep} = V_g + 1/(R_a + R_s) \qquad (11)$$


where $V_g$ is the gravitational settling velocity and the resistances to deposition are aerodynamic ($R_a$) and surface ($R_s$). The aerodynamic resistance is given as

$$R_a = (\ln (z_{ref}/z_0) - \psi_H)/(ku_*) \approx (\zeta_{ref} - \psi_H)/(ku_*)$$




where $z_0$ is a roughness length, presumed to be $z_{0m}$ and $\psi_H$ is a stability function from MOST. It is applied with $z_{ref}$ $\gg z_0$ and so one can use $\zeta_{ref} = \ln ((z_{ref} + z_0)/z_0)$. In neutral stratification $\psi_H = 0$ and for deposition to a water surface it is reasonable to set $R_s = 0$, unless it could be used to differentiate between $z_{0m}$ and $z_{0c}$. We can then write the relationship as


$$V_{dep} = V_g (1 + 1/(S\zeta_{ref})) \qquad (12)$$

From our CFLGS profile (Eq 8) we can derive an alternative expression for deposition velocity,

$$V_{dep} = F_{Qc}/Qc(z_{ref}) = V_g/(1-exp(-S\zeta_{ref})). \qquad (13)$$

This has similarities with the Zhang et al (2001) form. First we note that $V_{dep} \geq V_g$. For our over water situation with $R_s = 0$, for large $\zeta$, $V_{dep} \rightarrow V_g$ and also in the limit as $V_g \rightarrow 0$ both will have $V_{dep} \rightarrow ku_*/\zeta_{ref}$ when $R_s = 0$ and $z_{0m} = z_{0c}$, or if we set $R_s = ln(z_{0m}/z_{0c})/(ku_*)$. The Zhang et al (2001) and $z_{0c}$ approaches differ in detail between those limits and

an illustration is given in Section 4, Fig 3. The $\zeta_{ref} \rightarrow 0$ limit is similar in both approaches with $R_s = 0$ since then $V_{dep} \rightarrow \infty$ as $z \rightarrow 0$ and the aerodynamic resistance goes to 0.

There is little discussion of the variation of $V_{dep}$ with $z_{ref}$ in the literature, most of the focus being on variation with particle diameter (d). Farmer et al (2021) comment that "There are serious problems with our current understanding

of deposition rates", but provide (Fig 3 in the paper) a summary of observed, and some modelled, values of deposition rate over different types of surface (grassland, forest, water and cryosphere) for a range of particle diameters from 0.01 to 100 μm. Our main concerns are with fog and other aerosol with diameters in the 0.5 to 50 μm range and their deposition to water surfaces. Farmer et al's plot (Fig 3c) shows an approximate $V_{dep} \sim d^2$ relationship, but with $V_{dep} > V_g$. For more general aerosol the particle density and shape will modify $V_g$ and $V_{dep}$ and cause some

of the scatter, along with variations in $u_*$ and $z_{ref}$. Sehmel and Sutter (1974) report on wind tunnel determinations of deposition velocity over water. Their Figure 3 results for uranine particles (density 1500 kg m$^{-3}$) shows results at low wind speeds with $V_{dep}/V_g \sim 1$, while at higher wind speeds and for diameters in the range 1-30 μm have $V_{dep}/V_g$ increasing from about 3 to about 10.

**4. Some profiles**

The expected values of $V_g$ and $u_*$ should be considered. Fog droplets have a range of sizes but most fall in the diameter range 0-50 μm, often with bimodal distributions and peaks around 6 and 25 μm (see for example Isaac et al, 2020). Applying Stokes law with appropriate values for water droplets (see Rogers and Yau, 1976) for these peak sizes we get $V_g$ values of 0.0011 and 0.0192 m s$^{-1}$. These terminal velocities are clearly small compared to wind

speed but for the larger diameter droplets, where the bulk of the liquid water content, $LWC (=\rho_a Qc)$, is often measured, the terminal velocity corresponds to 69 m per hour and will represent a considerable removal rate in fog which may last several hours or days. The key parameter in our constant flux with gravitational settling model is $S =$





$V_g/ku$. In moderate winds over the ocean one might expect $u_*$ values in the 0.15-0.6 m s$^{-1}$ range, while in radiation

fog in light winds over land it could be lower. The parameter, $S$ will thus generally be in the range 0.0 to 0.3 over

water but could be unlimited over land.

At low values of $S$ gravitational settling will have low impact and $Qc$ profiles will be approximately logarithmic. To

illustrate this Fig. 1 shows $Qc$ constant flux profiles with linear and log vertical axes and a range of $S$ values. We

have scaled $Qc$ with a value at 50m. The main unknown is the value of $z_{0c}$. Here we use our first guess value ($z_{0c} =$

0.01m) indicating relatively efficient capture of water droplets by the water surface. These calculations are for

uniform sized droplets. Note that with high $S$ ($=V_g/ku_*$) values, maybe occurring with low $u_*$ and minimal

turbulence, the limiting case would be constant $Qc$ down to $z = 0$ and a discontinuity to $Qc = 0$ at the surface.

Calculations with $S = 1$ and 5 (not shown) confirm this.

One way to look at the relative importance of gravitational settling for these uniform size droplets is to consider the

relative contributions to the total downward flux of water droplets ($u_*q_{c*}$). The gravitational contribution is simply

$V_gQc$ while the turbulent diffusion contribution is,

$$ku_*dQc/d\zeta = u_*q_{c*}e^{-S\zeta}, \text{ where } \zeta = \ln\left((z+z_{0c})/z_{0c}\right) \tag{14}$$


The ratios of turbulent transfer (TT)/total flux and gravitational settling (GS)/total flux then become

$$TT = e^{-S\zeta} \quad \text{and} \quad GS = 1 - e^{-S\zeta} \tag{15}$$

Noting that $\zeta = \ln\left((z+z_{0c})/z_{0c}\right)$ we can see that these ratios depend on both $z_{0c}$, through the $z(\zeta)$ relationship, and $S$

and will vary with $z$. Fig. 2 illustrates this. It is important to note that Fig. 2 is based on our relatively low estimate

for $z_{0c}$, (0.01 m). If we increase it to $z_{0c} = 0.1$ m then turbulent fluxes become more important. We can see that the

$TT$ ratio is formally 1 at the surface, where $Qc = 0$ so there is no gravitational component. For very large $\zeta$ the $TT$

term would decay to 0 but this would be well above the constant flux layer approximation. At 50 m the value will

depend on $S$ and $z_{0c}$.

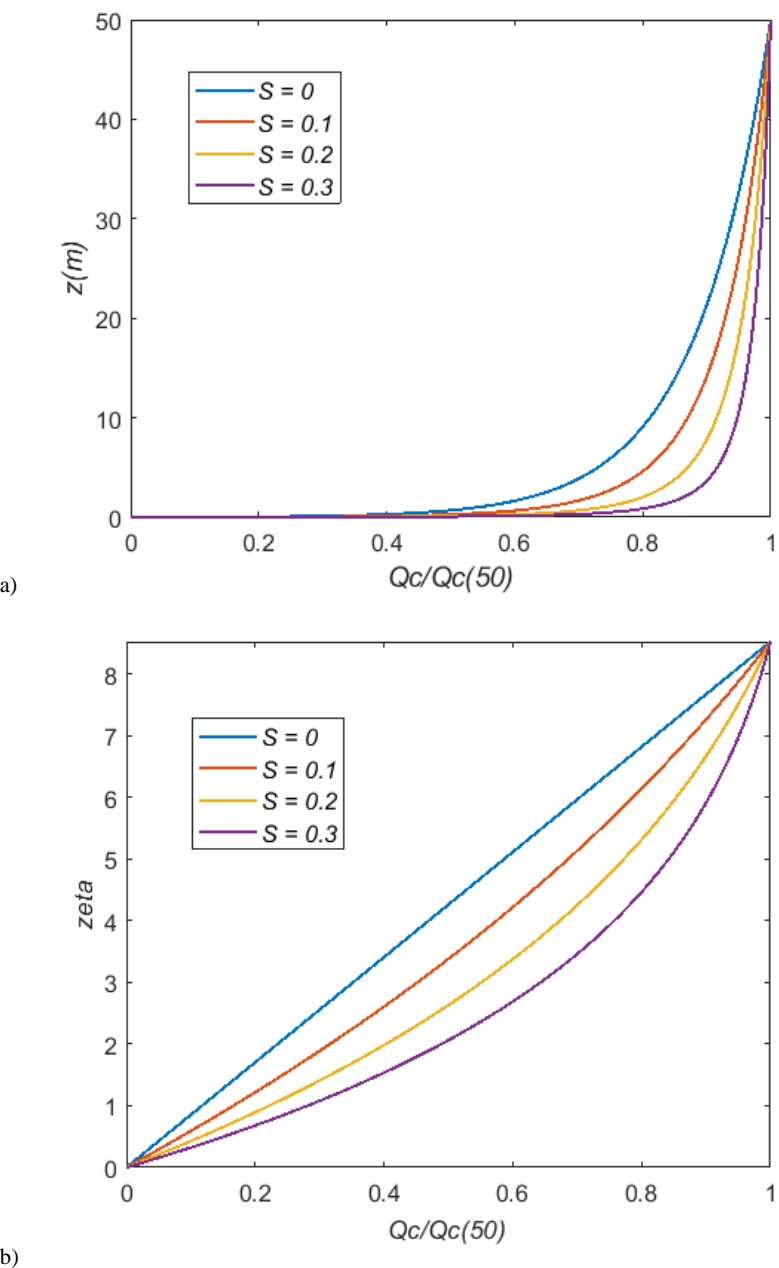

Fig. 1 *Qc* profiles, scaled by 50 m value, from surface to $z = 50$ m in constant flux layers with gravitational settling.

The surface roughness length for water droplet removal, $z_{0c} = 0.01$ m. Linear (a) and logarithmic (b) height scales.

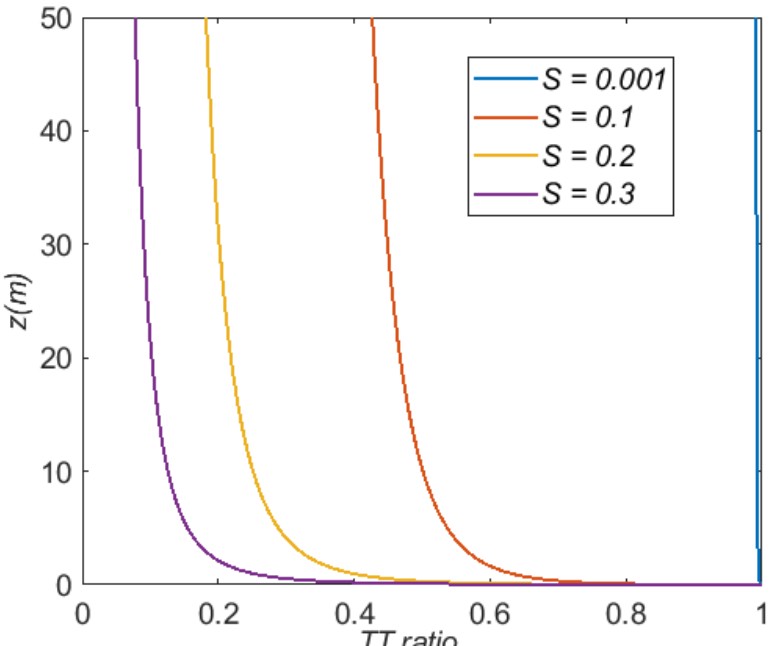

**Fig. 2** Variation of the Turbulent Transfer fraction of the total $Qc$ flux and its variation with $z$ and $S$. Note that these

$z$ values are based on a) $z_{0c} = 0.01$ m

We can also use Equations (12) and (13) to compute deposition velocities arising from the combination of

gravitational settling and, in Zhang et al's (2001) dry deposition terminology, aerodynamic resistance, although we

use $z_{0c}$ rather than $z_{0m}$ in the expression for $R_a$. Results in Fig 3 show similar variations with S, but note we are using

log scales for $V_{dep}/V_g$ and for $z_{ref}$.

With $z_{0c} = 0.01$ m and $\zeta = \ln ((z+z_{0c})/z_{0c})$ note that $z = 50$ m corresponds to $\zeta = 8.517$ while $\zeta = 4$ is only $z = 0.546$ m

and $\zeta = 6$ is $z = 4.03$m. There are differences with the Zhang et al (2001) formulation giving higher $V_{dep}/V_g$ estimates

than CFLGS, especially for the higher values of S in the $\zeta_{ref} > 6$ , $z_{ref} > 4$ m range. Both show dependence on $\zeta_{ref}$,

which is rarely commented on when deposition velocity values are reported, the emphasis being placed on aerosol

diameter as in Farmer et al's (2021) figures and tables. For aerosols in general we need better determination of

deposition velocity, $V_{dep}$, over all surfaces. Based on the analysis presented here it could be argued that more

attention should be paid to the parameter $S = V_g/(ku_*)$ and to the height $z_{ref}$ at which $V_{dep}$ can be applied.




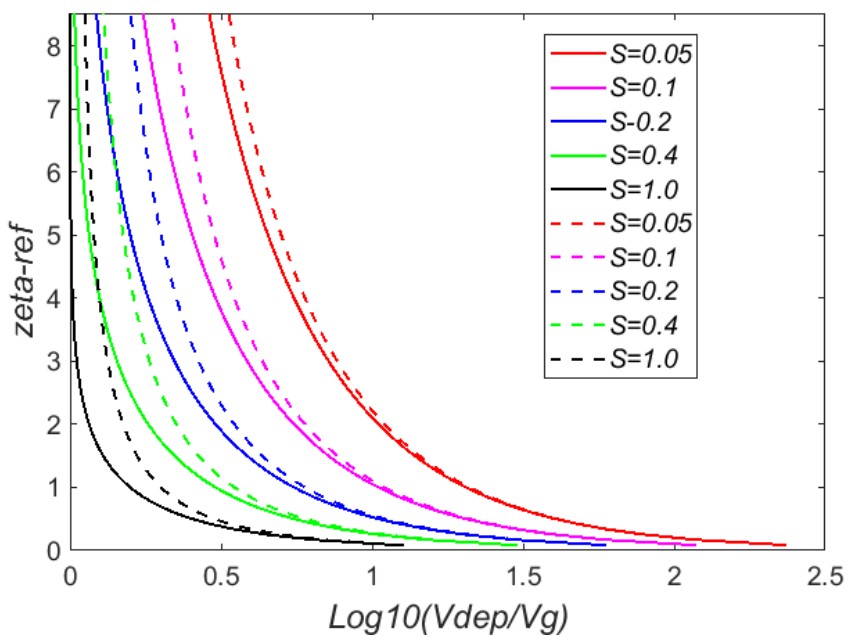


**Fig3**. Variations of deposition velocity $V_{dep}/V_g$ with $\zeta_{ref}$ and $S$. $z_{0c} = 0.01$ m. Solid lines are based on CFLGS (Eq 13) and dashed lines are Zhang et al's (2001) model with $R_s = 0$ and $R_a$ (Eq 12) as discussed in the text.

**4. Stable Stratification Case**

Over land, radiation fog often occurs at low wind speeds with stable stratification. For constant flux boundary layers in these circumstances MOST has, for velocity, $K_m = k(z+z_{0m})/ \Phi_M(z/L)$ and

$$\Phi_M(z/L) = 1 + \beta (z+z_{0m})/L : U = (u_*/k) (ln ((z + z_{0m})/z_{0m}) + \beta z/L). \qquad (16)$$


Observed profiles give $\beta = 5$ (Garratt 1992, p52). If we extend this idea to $K_{Qc} = k(z+z_{0c})/ \Phi_{Qc}(z/L)$ with a similar form for $\Phi_{Qc}$ we need to solve,

$$V_g Qc + [ku_* (z + z_{0c})/ \Phi_{Qc}(z/L)] dQc/dz = F_{Qc} = u_* q_{c*},$$

or

$$dQc/dz + S\{(1+\beta (z+z_{0c})/L)/(z+z_{0c})\}Qc=(q_{c*}/k)(1+\beta(z+z_{0c})/L)/(z+z_{0c}); \quad S=V_g/(ku_*)$$

The Integrating Factor is $exp( \int S(1/(z+z_{0c})+\beta/L)dz = (z+z_{0c})^S exp(S\beta z/L)$ so that

$$d [(z+z_{0c})^S exp(S\beta z/L)Qc] /dz = (q_{c*}/k)(1+\beta(z+z_{0c})/L) (z+z_{0c})^{S-1} exp(S\beta z/L) \qquad (17)$$




and we need to integrate the RHS. To do this it is convenient to let $\beta(z+z_{0c})/L = x$ and the integral that we need is of

$$(q_{c*}/k)(L/\beta)^{S-1}exp(-Sx_0) \; \{(1+x)x^{S-1}exp(Sx)\}, \qquad \text{where } x_0 = \beta z_{0c}/L \qquad (18)$$


After some guidance and a few trials one can see that $d/dx\{x^S exp(Sx)\} = (Sx^{S-1} + Sx^S)exp(Sx)$ and the integral required is simply $F(x,S) = x^S exp(Sx)/S$. We then evaluate $F(x,S)$ at $z = 0$, $x = \beta z_{0c}/L$ and any other $z$ to allow us to plot $Qc$ profiles. With stable stratification and light winds the constant flux approximation would only apply to a relatively shallow layer so we normalize with $Qc(ztop)$ and set $ztop = 20$ m in these cases. If $Qc = 0$ at $z = 0$ we then have,


$$Qc(z) = (q_{c*}/k)(L/\beta)^{-1}exp(Sx_0) \; [exp(-Sx) \; x^S)] \; [F(x,S) - F(x_0,S)], \qquad (19)$$

and we can then plot the ratio $Qc(z)/Qc(ztop)$ as in Fig. 4. For $S = 0$, with no gravitational settling, the profile will be essentially the same as the velocity profile in (A1) above, i.e.


$$Qc(z) = (q_{c*}/k) \; (ln \; ((z + z_{0c})/z_{0c}) + \beta z/L). \qquad (20)$$

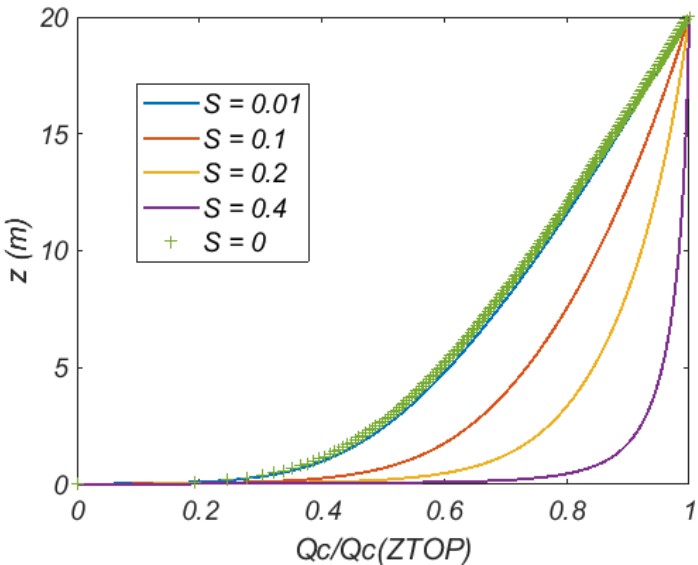

**Fig 4**. Qc/Qc(ztop) profiles with stable stratification, assuming $\Phi_{Qc}(z/L) = 1 + \beta(z+z_{0c})/L$. We set $\beta = 5$, $L = 20$m
and $z_{0c} = 0.01$m.

In addition to $z_{0c}$ and S the key parameter is the Obukhov length, $L = -\rho c_p u_*^3\theta/(kgH)$, (>0). Neutral stratification corresponds to $L \rightarrow \infty$ while stable stratification relationships ($H < 0$, $L > 0$) are generally limited to $0 < z/L < 1$. If we are concerned with height ranges up to 10 or 20m then $L = 10$m would be considered as a very low value maybe with
$u_* \approx 0.13$ ms$^{-1}$ and $H \approx -20$ Wm$^{-2}$ as possible values. Figure 4 shows Qc(z)/Qc(20m) profiles in a typical case with our





standard value, $z_{0c}$ = 0.01m. We set L = 20m and use a range of S values. For large droplets, S = 0.4, Qc flux is dominated by gravitational settling and reductions in Qc towards 0 only occur in the lowest few m. For smaller particles, S = 0, 0.01, 0.1 turbulent mixing dominates the deposition process. Note that the $S = 0$ points (log + linear profiles) and the $S = 0.01$ line, almost overlap as one confirmation of solution form.


## 5. Conclusions and Suggestions

The basic idea behind this analysis was that, in marine fog, cloud droplets can both fall toward the underlying surface through gravitational settling and be diffused towards the surface by turbulence and on contact they can coalesce with an underlying water surface. Taylor et al (2021) apply these ideas to fog modelling with the WRF

model. During reviews of that work, and an earlier version of the current paper, it became clear that some reviewers were reluctant to accept that turbulence could cause fog droplets to collide and coalesce with an underlying water surface, and even more reluctant to see this as a constant flux layer situation. Fog droplets are perhaps a special case in that there could be fluctuations in relative humidity allowing transfers between water droplets and water vapour, and variations of droplet size. It can still be argued that our conceptual model of fog droplets and cloud liquid water

being generated near the top of a fog layer, perhaps as a result of radiative cooling is useful. Once created the droplets can travel downward via both gravitational settling and turbulent diffusion towards a sink at the water surface. If the relative humidity is at 100% throughout this descent it seems reasonable assume a constant flux layer.

The same constant flux layer concept can apply in the case of other aerosols, provided that they are inert and without

sources or sinks in the air. Desert dusts, various pollutants or micro-plastic fragments being blown out over lakes or the sea from sources on land are examples. Here we could anticipate a situation with initial mixing through a relatively deep atmospheric layer over land with minimal deposition being advected over an aerosol capturing water surface so that one could envisage a situation over the water with a constant downward flux of aerosol due to gravitational settling plus turbulent diffusion in a low level constant flux layer.


In considering aerosol the recent review of dry deposition by Farmer et al (2021) and the widely used scheme of Zhang et al (2001) clearly show us that deposition velocity frequently exceeds gravitational settling velocity, especially over water. This seems to be readily accepted in the atmospheric chemistry community with models developed such as Eqs (10-12) above, and also for fog deposition to vegetation (Katata, 2014). One can use these

ideas in modelling work, adapting the approach of Katata et al (2010, 2011) for radiation fog over forests. This is the approach adopted in Taylor et al (2021) to deal with marine advection fog over the ocean. A critical unknown parameter in this work is the deposition velocity relating $Qc$ at the lowest model level to the downward flux to the surface due to turbulent transfer. As in the analysis above, one can use a roughness length for cloud droplets, $z_{0c}$, as a tuning parameter when suitable Qc profile measurements are available.


The bottom line is that this removal process needs to be taken account of in modelling and forecasting fog occurrence and development and we need to know more about it. Fog is an intermittent phenomenon so setting up





50-m or higher measurement masts in fog-prone locations will be good start. The PARISFOG study (Haeffelin et al, 2000) included 30-m masts and LANFEX (Price et al, 2018) used 50-m masts but the profile measurements did not

include fog water, Qc, or visibility. In-situ vertical profiles of Qc were also missing in field programs like FRAM (Gultepe et al, 2009) and C-Fog (Fernando et al, 2021). C-Fog instrumentation at various sites included 10-m and 15-m masts and also a Radiometrics microwave radiometer for Qc profile measurements. These may well report interesting measurements but better vertical resolution is desirable. There were Qc measurements at two or more levels in earlier field measurements reported by Pinnick et al (1978) and Kunkel (1984) showing increases with

height. More such measurements are needed with multiple measurement levels and measuring droplet size distributions, $Qc$ or $LWC$ values and ideally $Qc$ fluxes, along with wind, turbulence, temperature and humidity profiles plus surface pressure and fluxes of momentum, heat and water vapour. Visibility measurements at multiple levels, 4 component radiation and air, aerosol and fog chemistry measurements could also play an important role in fog. From the modelling perspective we need values for $z_{0c}$, which will depend on surface type and, on droplet

diameter and on wind speed or friction velocity. Assuming that the lower layers, say 10-30 m of a deep fog layer, are in a relatively steady, constant flux layer situation then the CFLGS profiles developed above could provide a framework for analysis of fogs and the improvement of fog models.

**Acknowledgements** Financial support for this research has come through a Canadian NSERC Collaborative Research and Development grant program (High Resolution Modelling of Weather over the Grand Banks) with

Wood Environmental and Infrastructure Solutions as the industrial partner. Discussions with Anton Beljaars, George Isaac and York colleagues over the past year have led me to some of the ideas behind this paper.

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





**Code/Data Availability**

390 Calculations were made with simple Matlab code, maybe 20 lines for each figure. They can be made available if
needed.

**Author Contibution**

This is independent work by the single author.

395

**Competing Interests**

None.