# Peer review of "Constant Flux Layers with Gravitational Settling: with links to aerosols, fog and deposition velocities."

_Atmospheric Chemistry and Physics, 2021_

## Author Response (AR1)

Responses to RC1 and RC2 were posted on the ACP web site on 11 October

Since then I have gone deeper into "deposition velocity" issues and have now expanded my discussion on that topic. This is somewhat contrary to my earlier statement in **AC2**: https://acp.copernicus.org/preprints/acp-2021-594/#discussion, Peter A. Taylor, 11 Oct 2021, which read,

" In view of this I will limit the discussion of deposition velocity, $V_{dep}$, in the revised version of the present paper to pointing out the dependence of $V_{dep}$ on the reference height, $z_{ref}$, and the friction velocity, $u*$, which often seems to be overlooked. The constant flux layer Qc profiles in Figures 1 and 4 can be simply inverted to show $V_{dep}(z)/V_{dep}(50) = 1/[Qc/Qc(50)]$ to illustrate this point, and can be discussed in sections 2 and 4 (Some Profiles). Section 3 (Dry Deposition Velocities) and Figure 3 will be removed. Note that the original version had a section numbering error (two sections 4), which will be corrected."

Rather that plot $V_{dep}(z)/V_{dep}(50)$ I realised that my CFLGS $V_{dep}(z)$ would differ from the Zhang et al version at 50 m and it is better not to scale by the 50 m value. The original Fig 3 has been removed, and also Fig 2 has become Fig 3. However a new Fig 2 now shows $V_{dep}$ values and compares with the Zhang/Slinn formulation. The revised paper in fact expands on the $V_{dep}$ discussion. In my view, this discussion demonstrates a weakness in the Zhang/Slinn formulation which effectively adds two effective deposition velocities together. In the CFLGS approach the two combine to give a slightly (20%) lower estimate for $V_{dep}$ than the, widely used, Zhang/Slinn formulation. This applies for surface deposition of any aerosol with significant gravitational settling contributions to $V_{dep}$.

Although 20% is small compared to other uncertainties in $V_{dep}$, I do consider that this is an important modification to the $V_{dep}$ literature and hope that these revisions will be acceptable. In view of this I have also broadened the focus a little and removed "over water" from the title and at several points in the text. I also just found some earlier work (Csanaday, 1973, Venkatram and Pleim, 1999, and notes in Giardina and Buffa, 2018) which proposed essentially the same CFLGS approach as in the present paper, but with less detail. These are now referenced.

RC2 is concerned about the assumption that "surface resistance", $R_s = 0$, and considered that it should always be > 0. This is a reasonable argument with the $R_s$ formulation given by Zhang et al (2021, Eq 5) but the definition of total resistance, $R_a + R_s$ involves the roughness length for momentum, $z_{0m}$ which may not be well determined. With the formulation used here, following Garratt (1992, Section 3.3.3), $R_s = (ku*)^{-1}\ln(z_{0m}/z_{0c})$, $R_s$ could be positive, maybe most common, zero or negative, if for example we were looking at water droplets colliding and coalescing with water, or a wet, hydrophilic surface. In our deposition velocity discussion, Fig 2 and the discussion includes cases with $ku*R_s = +/-$ 2.3. It can be seen that for $R_s > 0$ the deposition velocity variations are more uniform with height, but still exhibit similar CFLGS versus Zhang/Slinn differences as the $R_s = 0$ case.

Bottom line for me is that, as Farmer et al (2021) point out, there is much uncertainty in $V_{dep}$ measurements and separating out $R_a$ and $R_s$ contributions, or in my context, determining $z_{0m}$ and $z_{0c}$, is very difficult and needs more, and very careful, observation. One step in that direction will be a better combination of gravitational settling and turbulent fluxes, which is my goal in the present paper. Note that the discussion in Sections 2 and 4 has no discussion of $z_{0m}$ or $R_s$, it just arises in section 3.

The ==detailed points raised by both reviewers== have been addressed,

RC1

1. ==Line 240 ;Reference is made to radiation fog over land as an example of fog with stable stratification. It would be good to mention advection fog over water where fog forms in a transition from warm to cold SST.==

I have made this amendment, lines 258,9 in the revised paper.

RC2

Introduction provides a good discussion on the history of the similarity theory, which is based on constant flux layer situations in steady state. Cases under neutral stratification were discussed. May be the author can also provide a brief discussion on how the theory was expanded to unstable stratification, for a complete picture on this topic. The last paragraph in Introduction may be reorganized a bit so the readers can easily find out what materials are from existing theory, what are to be proposed in this study, the major goals of this study, and/or an outline of the following sections.

I added (Lines 73-76 ) " The main innovation in this short communication will be to combine the effects of turbulent transfer towards an underlying surface with gravitational settling ($V_g$). This is done in a similar way to that proposed by Venkatram and Pleim (1999) and differs from the additive deposition velocity form used by Zhang et al (2001) and Slinn (1982). The parameter, $S = V_g/ku_*$ plays a key role."

1. Line 125: delete "(i.e. not involving rain or snow - wet deposition)" to avoid confusion. This is because dry deposition happens all the time, even during precipitation events. As long as the pollutants are not incorporated into hydrometers before being adsorbed by surface, this process is referred to as dry deposition.

Agreed and comment deleted.

2. Line 130: add particle sizes for each scenario: "If gravitational settling is the main cause of $F_{Qc}$ (e.g., for particles large than several micrometer), we would expect little change in Qc with height, but if turbulent transfer is dominant (e.g., for very small particles) then the choice of $z_{ref}$ could be important"

Text has been moved and changed and a size range (1-20 μm) indicated on line 151.

3. Line 148: This is my biggest concern of this study. Is this Rs defined here the same meaning as that in Zhang et al. (2001)? If so, then Rs cannot be assumed to be 0. In any particle dry deposition model, Rs can never be 0, and actually is very large over water surface (Rs is usually >> Ra under unstable and neutral stratification, and on a similar magnitude to Ra under stable stratification).
Because all of the following sections are based on this assumption (Rs=0), which I do not agree with, I do not have much confidence on all the results generated here. If not setting Rs as 0, is there a way to still generate some analytical formula? I guess not. If section 3 was based on a false assumption (i.e., Rs=0), and if the author cannot fix the error, then this section should be deleted from this manuscript. To make the study still publishable, the author can change the study to a "Short communication" and then focus on Section 2 only. If possible, expand Section 2 to cover all stratification conditions over water surface, and if possible, provide some recommendation on how to expand to other land surfaces (smooth ones first and then vegetated surfaces).

The Rs issue is discussed above, and in the revised paper, lines 172 - 193. Deposition velocity calculations have now been added with non zero Rs. I am quite happy to have this paper considered as a "Short communication" but could not find details of this option on the ACP web site.

The stratification issue was addressed in https://acp.copernicus.org/preprints/acp-2021-594/#discussion which said,

"The suggested extension to cover all stratifications could be done via numerical solution of the equation on line 249 but, in unstable stratification (L < 0), it is not clear what an appropriate stability function, $\Phi_{Qc}(z/L)$ should be. For stable stratification it is generally accepted that $\Phi_H = \Phi_M$ and an extension to use the same equation for $\Phi_{Qc}$ seems reasonable. For unstable stratification $\Phi_H \neq \Phi_M$ and it is not clear what should be used for $\Phi_{Qc}$. In addition, it should be noted that in the stable case I could find an analytic solution, but that would be more difficult with the unstable case. In the advective marine fog situations which initiated this work we were only concerned with warm air over cold water and so focussed on stable stratification."

Comments on the unstable case have been added in lines 316 - 322 of the revised manuscript.

---

## Author Response (AR2)

Response to referees

Title: Constant Flux Layers with Gravitational Settling: with links to aerosols, fog and deposition velocities
Author(s): Peter Allan Taylor
MS No.: acp-2021-594

The revised version was accepted. I appreciate the referee comments, and note the comment regarding canopy issues. I will bear that in mind for further work on this topic.